# Accurate Blood-Based Diagnostic Biosignatures for Alzheimer’s Disease via Automated Machine Learning

**DOI:** 10.3390/jcm9093016

**Published:** 2020-09-18

**Authors:** Makrina Karaglani, Krystallia Gourlia, Ioannis Tsamardinos, Ekaterini Chatzaki

**Affiliations:** 1Laboratory of Pharmacology, Medical School, Democritus University of Thrace, 68100 Alexandroupolis, Greece; mkaragla@med.duth.gr; 2Gnosis Data Analysis PC, Science and Technology Park of Crete, N. Plastira 100, GR-700 13 Vassilika Vouton, Greece; tsamard.it@gmail.com; 3Department of Computer Science, University of Crete, GR-700 13 Vassilika Vouton, Greece; krysgourlia@gmail.com; 4Institute of Applied and Computational Mathematics, Foundation for Research and Technology Hellas, GR-700 13 Vassilika Vouton, Greece; 5Institute of Agri-Food and Life Sciences, University Research Centre, Hellenic Mediterranean University, GR-71410 Heraklion, Greece

**Keywords:** Alzheimer’s disease, predictive model, machine learning, blood, classifier

## Abstract

Alzheimer’s disease (AD) is the most common form of neurodegenerative dementia and its timely diagnosis remains a major challenge in biomarker discovery. In the present study, we analyzed publicly available high-throughput low-sample -omics datasets from studies in AD blood, by the AutoML technology Just Add Data Bio (JADBIO), to construct accurate predictive models for use as diagnostic biosignatures. Considering data from AD patients and age–sex matched cognitively healthy individuals, we produced three best performing diagnostic biosignatures specific for the presence of AD: A. A 506-feature transcriptomic dataset from 48 AD and 22 controls led to a miRNA-based biosignature via Support Vector Machines with three miRNA predictors (AUC 0.975 (0.906, 1.000)), B. A 38,327-feature transcriptomic dataset from 134 AD and 100 controls led to six mRNA-based statistically equivalent signatures via Classification Random Forests with 25 mRNA predictors (AUC 0.846 (0.778, 0.905)) and C. A 9483-feature proteomic dataset from 25 AD and 37 controls led to a protein-based biosignature via Ridge Logistic Regression with seven protein predictors (AUC 0.921 (0.849, 0.972)). These performance metrics were also validated through the JADBIO pipeline confirming stability. In conclusion, using the automated machine learning tool JADBIO, we produced accurate predictive biosignatures extrapolating available low sample -omics data. These results offer options for minimally invasive blood-based diagnostic tests for AD, awaiting clinical validation based on respective laboratory assays. They also highlight the value of AutoML in biomarker discovery.

## 1. Introduction

Alzheimer’s disease (AD) is the most common form of neurodegenerative dementia [1] and the fifth leading cause of death for those over 65 years [2]. AD is considered to be a chronic and slowly progressive disease, with a slow buildup of pathology during a long pre-symptomatic period [3]. AD causes a progressive loss of cognitive abilities, short-term memory impairment being the most typical initial symptom. However, there are also atypical clinical presentations of AD, e.g., primary progressive aphasia or posterior cortical atrophy. Medical diagnosis of AD is difficult, particularly at the early stage of the disease, mainly because symptoms are often dismissed as normal consequences of aging [4]. As of today, clinical diagnosis of AD is usually based on medical records, physical, and neurological examination, neuroimaging, laboratory tests, and neuropsychological evaluation, while its final diagnosis can only be achieved by autopsy [4], making the discovery of early and accessible biomarkers a major challenge. Early AD diagnosis in middle-aged individuals years ahead of cognitive decline might be the key to successful treatment, as currently available therapeutics only allow small benefits for diagnosed AD patients.

Targeted discovery approaches have been previously employed to identify some initial biofluid markers for AD diagnosis. Cerebrospinal fluid levels of amyloid beta protein (Aβ) and the microtubule-associated protein tau display good diagnostic utility and can be used to monitor aspects of therapeutic development [3]. However, there are no reliable blood biomarkers, although there is an emerging literature on P-tau concentration in neuronally derived blood exosomes, with contrasting results with regard to association with AD clinical manifestations [5]. The benefit of searching for blood-based biomarkers is evident due to the easy and minimally-invasive nature of blood sample collection compared with any other biofluid. Blood transcriptomics, proteomics, and/or metabolomics may pave the way for the development of accurate, cost-effective, and minimally-invasive AD diagnostics and/or screening tools for high risk individuals.

In the present study, we utilized publicly available multi -omics datasets from studies in AD blood for automated machine learning (autoML), using the Just Add Data Bio (JADBIO) platform [6] JADBIO is a recently launched autoML technology readily applicable to low-sample, high-dimensional data producing accurate predictive models with their corresponding biosignatures [7,8,9,10,11]. For a given outcome and set of features, JADBIO provides both (i) the minimal subset of predictors (namely, the biomarker biosignature) that ensures maximal predictive power and (ii) the optimal predictive model associated with the selected biosignature. This is achieved by combining the feature selection method Statistically Equivalent biosignatures (SES) [12] with Support Vector Machines, Random Forest, and Penalized Linear Models algorithms. Cross validation is used in order to optimize the algorithms over their corresponding hyper-parameter configurations and to select the best performing configuration. The expected predictive performance on external data is estimated using bootstrapping techniques and was proven free of over-optimistic estimates (Bootstrap Bias Corrected cross validation (CV) protocol, BBC-CV [13]).

Using JADBIO, we reprocessed publicly available datasets against clinically diagnosed AD to produce accurate biosignatures/models that could subsequently be translated into cost-effective benchmark diagnostic solutions.

## 2. Materials and Methods

### 2.1. Datasets

In the present study, we searched for publicly available -omic datasets generated from blood samples of AD patients and cognitively healthy individuals that include both molecular profiles (transcriptomic, proteomic, and metabolomic), and curated meta-data (i.e., study design information and clinical data) in BioDataome [14], Metabolomics Workbench [15], and Gene Expression Omnibus (GEO) [16] repositories. BioDataome is an online repository with transcriptome (both microarray and RNA-seq) and epigenetics (methylation array) datasets, which uniformly processes and automatically annotates datasets from the GEO and the RECOUNT database [17]. The available information about the patients/cases in the datasets used regarding age and sex are presented in Appendix A.

### 2.2. Construction of Biosignatures/Models via AutoML

For automated machine learning we employed the autoML JADBIO [6] platform, version 1.1.20 (10 June 2020). Specifically, JADBIO [6] has the following functionality and properties: (a) given a 2D matrix of data, it automatically produces predictive models for a categorical (classification), continuous (regression), or time-to-event (survival analysis) outcome. No selection of appropriate algorithms to apply is necessary or tuning of their hyper-parameter values. Available classification algorithms are: Classification Random Forests, Support Vector Machines (SVM), Ridge Logistic Regression and Classification Decision Trees. (b) It identifies multiple equivalent biosignatures, (c) it produces conservative predictive performance estimates and corresponding confidence intervals. It reliably processes up to hundreds of thousands of features and sample sizes as low as a couple of dozen. JADBIO also employs the recently developed BBC-CV protocol for tuning the hyper-parameters of algorithms while estimating performance and adjusting for multiple tries. JADBIO architecture can be found in Montesanto et al., 2020 [7].

Seven different omics 2D csv matrixes, including molecular profile data and respective clinical information were uploaded in JADBIO and were automatically analyzed via extensive tuning effort. For all datasets the performance was evaluated via internal validation (BBC-CV within each dataset). In addition, external validation was plausible in the large-sample datasets (transcriptomic 2, 3 and 4), which were randomly automatically split into train and test sub-datasets by 70/30 ratio via JADBIO.

### 2.3. Correlation of Selected Features to AD

The biological involvement and related pathways of identified features, i.e., proteins, mRNAs and miRNAs to AD was searched using the GeneCards—The Human gene database tool (https://www.genecards.org/). MiRNA Predicted targets were identified in the miRbase database (http://www.mirbase.org/).

## 3. Results

### 3.1. Datasets

Searching in the available repositories, we were able to identify two metabolomic datasets, two proteomic datasets, three microRNA (miRNA) transcriptomic datasets (one excluded from further analysis due to very low sample number) and seven mRNA transcriptomic datasets (three excluded from further analysis due to very low sample numbers, non-availability of processed data or inclusion of only female samples). Specifically, AD patient samples and age and sex matched cognitively healthy individual samples were retrieved from the following datasets: the metabolomic datasets ST000046 (Project doi:10.21228/M88G6G) (metabolomic 1) and ST000433 Project DOI:10.21228/M8HS40) (metabolomic 2) from Metabolomics Workbench, the proteomic datasets GSE39087 [18] and GSE29676 [19] from GEO, pooled as they contained the same profiles (proteomic), the two miRNA transcriptomic datasets GSE46579 [20] (transcriptomic 1) and GSE120584 [21] (transcriptomic 2) from Bio-dataome and GEO respectively, and two mRNA transcriptomic datasets GSE63060 [22] (transcriptomic 3) and GSE63063 [22] (transcriptomic 4) from GEO. All in all, we analyzed seven different datasets in our study. Table 1 gives a summary of the dataset dimensions used in the study.

### 3.2. Biosignatures

Considering all the suitable multi-omics blood-based data from AD patients identified in the available depositories and age–sex matched cognitively healthy individuals, we produced biosignatures of AUC ranging from 0.489 to 0.975 described below.

#### 3.2.1. Proteomic Biosignature

AutoML analysis of the 9483-feature proteomic data from 25 AD patients and 37 cognitively healthy individuals led to one protein-based biosignature via a Ridge Logistic Regression model with seven protein predictors with high AUC (0.921 (0.849, 0.972)). The biosignature’s performance, selected predictors and model inspection are presented in Figure 1.

Selected protein features include: Leucine Rich Repeats and IQ Motif Containing Protein 2 (LRRIQ2), Calcium signal-modulating cyclophilin ligand (CAMLG), interleukin 4 (IL4), tropomyosin 1 (TPM1), interleukin 20 (IL20), diablo homolog (Drosophila) (DIABLO), and Serine/threonine-protein kinase 3 (VRK3). Relation to AD according to GeneCards search (https://www.genecards.org/Search/Keyword?queryString=Alzheimer%27s&startPage=0&pageSize=-1), revealed some relation to AD for all but the first two proteins, IL4 being the one presenting the highest score (Appendix A).

#### 3.2.2. miRNA Biosignatures

AutoML analysis of the 506-feature miRNA-transcriptomic data from 48 AD patients and 22 cognitively healthy individuals led to one miRNA-based predictive biosignature via a Support Vector Machines model with three miRNA predictors with high AUC (0.975 (0.908, 1.000)). Biosignature performance, selected predictors and model inspection are presented in Figure 2.

Selection of a clinical cut-off threshold taking into consideration cost and disease prevalence is facilitated by the JADBIO ROC curve outcome, which contains all possible trade-offs between the False Positive Rate and the True Positive Rate, as well as the Precision-Recall Curve which contains all possible trade-offs between the Precision and Recall that are achievable by the model. JADBIO calculates statistics, performance metrics, and confidence intervals for 10 such points on these two curves (Figure 2A,B). Performance metrics (sensitivity, specificity, PPV, NVP, etc.) for three standard cut-off thresholds are included in Appendix A.

Further interactive peruse for cut-off thresholds and metrics for this analysis is provided in the link https://app.jadbio.com/share/3f050861-da6b-447a-b2b1-908c71d65c3d.

Relation of the three identified miRNAs to the AD according to GeneCards search (https://www.genecards.org/Search/Keyword?queryString=ALZHEIMER&pageSize=-1&startPage=0) revealed relation to the disease, with MIR29C related to the MicroRNAs in cancer and Metastatic brain tumor pathway presenting the highest score (Appendix A). MIR30D and MIR182 were identified in the miRNA targets in extracellular matrix ECM and membrane receptors and MicroRNAs in cancer and Alzheimer’s Disease pathways, respectively.

AutoML analysis of the 2566-feature miRNA-transcriptomic data from 301 AD patients and 288 cognitively healthy individuals led to an mRNA-based predictive biosignature via a Ridge Logistic Regression model with 25 mRNA predictors with AUC 0.797 (0.746, 0.845). The biosignature’s performance, selected predictors and model inspection are presented in Appendix A.

Most importantly, the size of this dataset allowed further internal automated validation, i.e., this same transcriptomic dataset was randomly split into train and test sub-datasets by a 70/30 ratio. The train 2566-feature miRNA-transcriptomic data from 232 AD patients and 200 cognitively healthy individuals led again to a similar but not identical mRNA-based predictive biosignature via a Ridge Logistic Regression model with 25 mRNA predictors with high AUC 0.812 (0.758, 0.862). A total of 14 of predictors were common with the original signature. While validated in the test 2566-feature mRNA-transcriptomic data from 84 AD patients and 73 cognitively healthy individuals, this biosignature showed AUC 0.807, verifying the stability of model’s performance. The biosignature’s performance, selected predictors and model inspection are presented in Appendix A and the performance of the validation dataset is shown in Appendix A.

#### 3.2.3. mRNA Biosignatures

AutoML analysis of the 38,327-feature mRNA-transcriptomic data from 134 AD patients and 100 cognitively healthy individuals led to six mRNA-based statistically equivalent predictive biosignatures via a Classification Random Forests model with 25 mRNA predictors presenting high AUC (0.846 (0.778, 0.905)). A total of 24 of 25 predictors were common in all six of them. The biosignatures’ performance, selected predictors for all six models and model inspection are presented in Figure 3.

Relation of the 30 identified mRNAs to AD according to GeneCards search (https://www.genecards.org/Search/Keyword?queryString=ALZHEIMER&pageSize=-1&startPage=0) revealed no or poor relation, with the exception of CHAT gene of Choline O-Acetyltransferase, related to the pathways of Neurotransmitter Release Cycle and Transmission across Chemical Synapses (Appendix A).

The size of this dataset also allowed internal validation and was automatically split into train and test sub-datasets by a 70/30 ratio. The train 38,327-feature mRNA-transcriptomic data from 100 AD patients and 73 cognitively healthy individuals led to an mRNA-based predictive biosignature via a Classification Random Forests model with 25 mRNA predictors with high AUC (0.822 (0.740, 0.894)). A total of 11 predictors were common to the original signature. When this biosignature was applied in the test sub-dataset from 34 AD patients and 27 cognitively healthy individuals, it showed AUC 0.824, verifying model’s performance stability. The biosignature’s performance, selected predictors and model inspection are presented in Figure 4 and performance of the validation dataset is shown in Figure 5.

AutoML analysis of the 32,053-feature mRNA-transcriptomic data from 126 AD patients and 131 cognitively healthy individuals led to two statistically equivalent mRNA-based predictive biosignatures via a Classification Random Forests model with 25 mRNA predictors with AUC 0.726 (0.644, 0.802), with 24 common features. The biosignature’s performance, selected predictors and model inspection are presented in Appendix A.

This same transcriptomic dataset was split into train and test sub-datasets by a 70/30 ratio. The train 32,053-feature mRNA-transcriptomic data from 100 AD patients and 90 cognitively healthy individuals led to four statistically equivalent mRNA-based predictive biosignatures via a Classification Random Forests model, with 25 mRNA predictors of AUC 0.728 (0.636, 0.817) and 11common predictors. The test data from 26 AD patients and 41 cognitively healthy individuals showed AUC 0.814, again verifying model’s performance stability. The biosignature’s performance, selected predictors and model inspection are presented in Appendix A and performance of validation dataset is shown in Appendix A.

#### 3.2.4. Metabolomic Biosignatures

AutoML analysis of the 3734-feature metabolomic data from 15 AD patients and 15 cognitively healthy individuals as well as the 25-feature metabolomic data from 18 AD patients and 21 cognitively healthy individuals data produced low performing biosignatures. Specifically, the first dataset led to a biosignature via a Support Vector Machines model with four metabolite predictors with poor AUC (0.556 (0.375, 0.709)). Similarly, the second lipidomic data led to a biosignature via Support Vector Machines model with one lipid predictor again of poor AUC (0.489 (0.294, 0.688)).

## 4. Discussion

Automated Machine Learning (AutoML) is a new reality in translational medicine and molecular biology, that promises to democratize data analysis to non-experts, drastically increase productivity, improve replicability of the statistical analysis, facilitate the interpretation of results, and shield against common methodological analysis pit-falls such as overfitting [6]. Most importantly, it to extract maximum information from laborious and expensive array examinations from precious and scarce clinical samples towards personalized clinical decisions and disease management. In this study, we employed the autoML technology JADBIO, in order to obtain easily and quickly accurate predictive models for AD disease, using all available archived proteomics, metabolomics and transcriptomics datasets. The JADBIO platform has been designed for non-experts to deliver high-quality predictive and diagnostic models, employing standard, best-practices and state-of-the-art statistical and machine learning methods. It identifies multiple (in case of biological redundancy) equivalent biosignatures of selective predictive features. It scales up to hundreds of thousands of features, which implies that it can simultaneously consider multi-omics, clinical, and epidemiological parameters. JADBIO scales down to tiny sample sizes (e.g., 40) often encountered with molecular biology data, in the sense that it still manages to provide accurate, non-optimistic estimates of maximum predictive performance. JADBIO is specifically devised for life scientists without knowledge of coding, advanced statistics or data analysis expertise. It shields against typical methodological pitfalls in data analysis that lead to overfitting and overestimating performance and therefore to misleading results [6]. Using this innovative but validated autoML tool, we were able to construct three best performing accurate diagnostic biosignatures from proteomics, miRNA-transcriptomics and mRNA-transcriptomics data via Ridge Logistic Regression, Support Vector Machines and Classification Random Forests models. All three biosignatures (and their statistically equivalent ones) showed high performance and the miRNA-transcriptomic biosignature reached the highest AUC (0.975 (0.906, 1.000)) with only three miRNA-predictors. These three miRNAs were also shown to be related to AD in a Genecards search, pointing into a biological role of the pathways involved in disease pathophysiology.

The bootstrapping technique used performs a correction to the estimation of out-of-sample performance of the final model. The correction (adjustment) is required because JADBIO tries thousands of machine learning pipelines to identify the best one that produces the optimal, final model. The correction is conceptually similar to the Bonferroni adjustment required for multiple hypothesis testing due to performing multiple tests. Intuitively, the selection process, which selects the best out of numerous pipelines is bootstrapped. This technique has been shown to produce conservative estimates of performance in massive evaluation experiments with general types of data [13] as well as hundreds of -omics data [23]. It has been used to produce several novel scientific results [9,10,11,12].

Blood-based biomarkers have emerged as promising minimally invasive AD diagnostic options [24] and machine learning has been employed for their development. In fact, multiple miRNA or mRNA (gene expression) biosignatures have previously been re-ported for early AD diagnosis. Specifically, Leidinger et al. produced a 12-miRNA biosignature via support vector machines that showed accuracy of 0.930 in diagnosing AD [20]. This same 506-feature miRNA dataset was analyzed here via JADBIO, resulting in a shorter biosignature of three predictors performing even better (AUC 0.975). Reducing the dimensions of a signature is a great advantage in terms of translatability to cost-effective assays with less need for multiplexing. Recently, Zhao et al. produced a 12-miRNA biosignature developed via a random forest algorithm that showed good accuracy (0.760) for AD diagnosis [25]. Similarly, Shigemizu et al. analyzing the same 2566-feature miRNA dataset used in our study, produced a 78-miRNA biosignature based on a supervised principal component analysis logistic regression method that showed accuracy 0.873 [21]. Our autoML approach led to a reduced size biosignature via a Ridge Logistic Regression model (25 predictors), and internal validation confirmed stability of its performance. Overall, earlier produced miRNA-biosignatures through machine learning, contain multiple predictors hampering their clinical adaptation. An additional advantage of our approach is the application of several machine learning algorithms at once automatically revealing the best.

Omics mRNA data have also been analyzed to produce diagnostic biosignatures. Patel et al. using an XGBoost classification algorithm reported two mRNA-biosignatures, a 57-mRNA biosignature for the discrimination between AD patients and healthy individuals with 0.450 AUC and an 89 mRNA-biosignature for the discrimination between AD patients and non-AD patients (consisted of either healthy individuals or patients with other diseases) with AUC 0.860 [26]. Lee and Lee produced three mRNA-biosignatures via logistic regression, L1-regularized logistic regression, random forest, support vector machine, and deep neural network using three independent datasets reaching AUC 0.657, 0.874, and 0.804, respectively. [27] Li et al., using the same 38,327-feature mRNA transcriptomic dataset of our study (containing all the available samples) and trying “manually” different ML approaches, including support vector machines, random forest and logistic Ridge Regression models, produced a three-mRNA biosignature which discriminated between AD patients and controls with 0.860 AUC [28]. AutoML analysis of this dataset here led to six statistically equivalent biosignatures via a Classification Random Forests model of similar performance.

To the best of our knowledge, to date only Ravetti et al. produced a five-protein biosignature via machine learning models for predicting clinical AD disease with 0.960 total accuracy using more than 20 different classifiers available in the widely-used Weka software package [29]. Whereas the original proteomic datasets [18,19] pooled to the one analyzed here, were originally used to study AD-specific autoantibodies in serum, our autoML analysis led to a seven-feature proteomic accurate biosignature for the diagnosis of AD, also highlighting the importance of this meta-analysis for maximal conclusion extraction from a given experimental observation. Selected protein features include: Leucine Rich Repeats and IQ Motif Containing Protein 2 (LRRIQ2), Calcium signal-modulating cyclophilin ligand (CAMLG), interleukin 4 (IL4), tropomyosin 1 (TPM1), interleukin 20 (IL20), diablo homolog (Drosophila) (DIABLO), and Serine/threonine-protein kinase 3 (VRK3). Interestingly, although the majority of them were found to be related to the AD disease according to our GeneCards—The Human gene database search their low score values were so far leading to their neglect from further attention in biological pathway and hypothesis-driven publications. Even more surprising was that the last three proteins have never been related to the AD disease before and this would certainly be interesting to further investigate. The deployment of our approach led to their surfacing, highlighting the offer of secondary data extrapolation and information digging approaches of machine learning analysis for translational purposes.

Although, metabolomic measurements act as a molecular fingerprint of disease progression, it is widely accepted that there is a degree of uncertainty in these datasets due to the limited overall coverage of the mass spectrometry (MS) methods. Combined with the inherent irreproducibility of the MS measurements, they may lead to challenges and difficulty in the development of accurate predictive biosignatures, also observed here in our metabolomic biosignatures. Interestingly, Stamate et al., using a 883-feature metabolomic dataset produced three biosignatures: one with AUC 0.850 (0.800–0.890) via Deep learning, a second of AUC 0.880 (0.860–0.890) via an XGBoost model and a last of AUC 0.850 (0.830–0.870) via an Random Forest model, respectively. However, all those biosignatures contained as much as 347 predictors [30].

A limitation worth mentioning of our study is the absence of external validation (cross dataset analysis) of the biosignatures, due to the lack of independent datasets run with common platforms. However, JADBIO implements internal cross-validation with the bootstrap corrected cross-validation (BBC-CV) algorithm [6] and it has been shown to produce accurate out-of-sample estimates of predictive performance. We also validated the developed signatures in 70/30 subsets, whenever the sample size of the dataset allowed it, and were able to show the stability of their performance. Another limitation is the low sample numbers of the proteomic and the metabolomic datasets, which is however overcome by the autoML tool used resulting in promising results at least in the proteome biomarkers. Overall, the high performance of our blood-based biosignatures paves the way for prospective clinical validation for timely identification of people with AD, in terms of early and accurate minimally invasive diagnosis.

The only available information about the patients/cases in the datasets available in the repositories, also in those used in this study, are age and sex. Clinical information is restricted to the presence of diagnosed AD or not (cognitive healthy controls). No information whatsoever is provided on the stage, duration since diagnosis, disease progression, moreover on the type of dementia. This is a most common situation in the deposited -omics datasets which undoubtfully raises an intrinsic limitation of these data-driven approaches. Even more, as reanalyzing and reevaluating archived data attracts gradually the interest of researchers, as new analyzing tools emerge, such as that used in our study, the quality and completeness of the data stored becomes a major issue and groups producing the original datasets are encouraged to share more information about their samples. This is strongly supported by studies such as this one, aiming to highlight the potential of these approaches.

The biosignatures built here can only differentiate between AD and healthy controls, as this was the only available clinical end-point. Our analysis cannot offer any additional diagnostic information against other clinically important endpoints, such as the type of dementia, disease progression and therapeutic outcome. Nevertheless, they offer a mature starting point for researchers working on developing diagnostic tools in AD, should they choose to further check their clinical performance in new and well described datasets, hopefully saving time and resources.

Upon further validation in complete datasets by groups with a special interest in AD, the biosignatures described could offer feasible solutions for routine blood-based laboratory tests that could be realized in any standard equipped diagnostic lab. This is the most significant merit of the present study for the scientific community, moving from the -omics multi-dimensional results to simpler classifiers. Reducing the dimensions of a signature is a great advantage in terms of translationability to cost-effective assays with less need for multiplexing. In specific, proteomic and miRNA biosignatures could be applied in a clinical setting due to their limited number of predictors. Proteomic biosignature could be used as a seven-protein ELISA assay, while miRNA biosignature as a three-plex RT-PCR assay measured on plasma samples, both technically feasible.

Combining proteomic, transcriptomic and metabolomic data in a multi-omic approach could theoretically be implemented for more effective data exploitation. However, any datasets jointly including more than one modality on the same sample set are rare, obviously due to high resource requirements. Alternatively, efforts to create computational methods that “translate” one platform and technology (e.g., transcriptomic) to another (e.g., proteomics) are at the moment the spearhead of bioinformatics development, with several labs (including ours) intensively working on such technologies, however still with less-than-optimal results [31]. Nevertheless, as far as it concerns applicability as an effective test in a clinical setting, biosignatures combining features of different nature (i.e., both nucleic acids and proteins) would be technically more complicated to implement in a single blood sample.

## 5. Conclusions

In conclusion, using the autoML JADBIO tool, we produced specific predictive biosignatures extrapolating available low sample multi-omics data. These results offer options for minimally invasive blood-based diagnostic tests, awaiting clinical validation based on respective laboratory assays for the classification of Alzheimer’s disease. They also highlight the value of autoML in biomarker discovery.

All three reported biosignatures showing high performance and AUC > 0.85 are worthy of further validation. Still, if we were to choose one as the most promising, we would propose this to be the miRNA-transcriptomic biosignature based on three apparent advantages: highest performance [AUC 0.975 (0.906, 1.000)], fewer features (three predictors), entities allowing easy multiplexing assays (miRNA).

A side-conclusion of this study yet quite important, is the following: revisiting and reanalyzing “old” datasets is not only potentially fruitful but scientifically necessary. The current mentality of life-scientists is often to discard public data as “used”, under the assumption that all discoveries to be made, are already published. However, as new and powerful statistical and computational methods are introduced, such as JADBIO and AutoML, new types of analyses become possible and new patterns and results become ripe for discovery [32]. It is not only computational methods that change; the analysis context is constantly changing. It is thus worth re-analyzing past datasets in the context of new datasets submitted [23] recent scientific publications, or newly added or revised knowledge in biological databases (e.g., updated pathways).

## Figures and Tables

**Figure 1 jcm-09-03016-f001:**
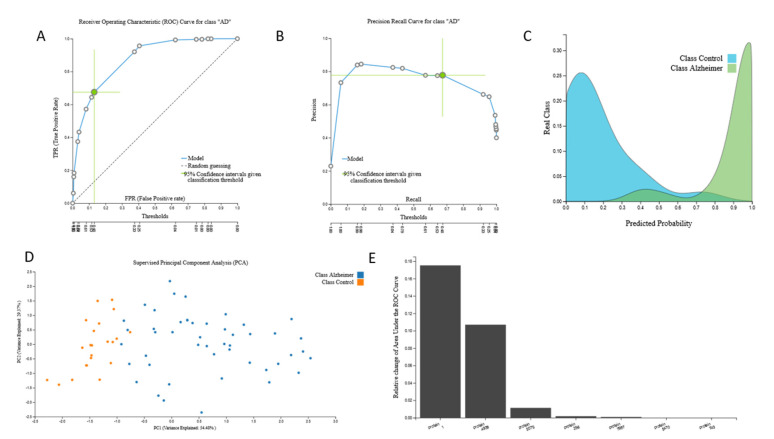
Proteomic signature. 9483-feature proteomic dataset signature performance, selected predictors and model inspection: (**A**). Receiver Operating Characteristic (ROC) Curve, (**B**). Precision-recall plot, (**C**). Probabilities density plot—These are the out-of-sample probabilities for a given sample to belong to Alzheimer’s disease (AD) class. (**D**). Supervised Principal Component Analysis plot depicting discrimination between AD patients and age–sex matched cognitively healthy individuals, in the space of the selected features. (**E**). Cumulative feature importance plot of the 7 protein predictors of the signature - y axis represents the Relative change of the Area under the ROC curve achieved by adding, one at a time, the features presented with that specific order.

**Figure 2 jcm-09-03016-f002:**
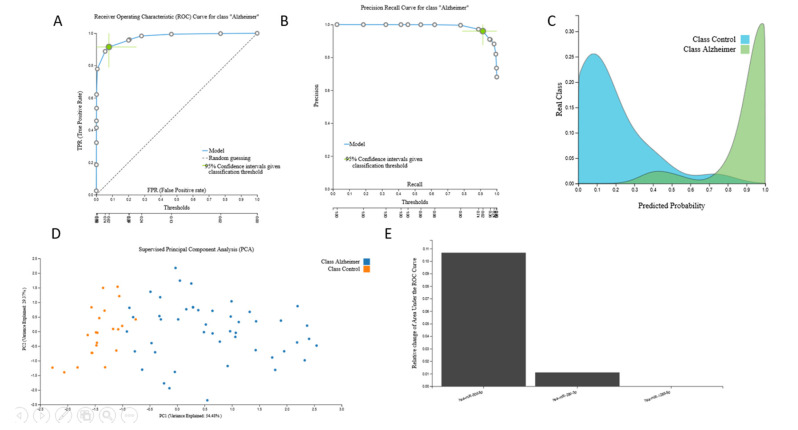
miRNA-transcriptomic signature. The 506-feature miRNA-transcriptomic dataset signature performance, selected predictors and model inspection: (**A**). Receiver Operating Characteristic (ROC) Curve for AD class, (**B**). Precision-recall plot, (**C**). Probabilities density plot, (**D**). Supervised Principal Component Analysis plot depicting discrimination between AD patients and age–sex matched cognitively healthy individuals, (**E**). Cumulative feature importance plot of the 3 miRNA predictors of the signature.

**Figure 3 jcm-09-03016-f003:**
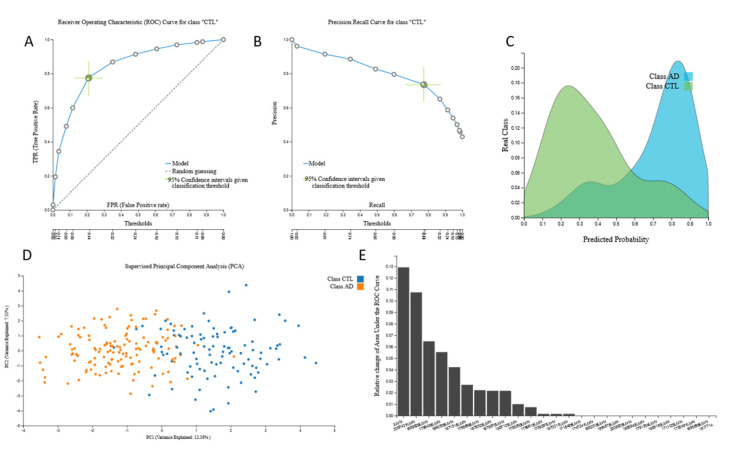
The 38,327-feature mRNA-transcriptomic original dataset signature performance, selected predictors and model inspection: (**A**). Receiver Operating Characteristic (ROC) Curve for AD class, (**B**). Precision-recall plot, (**C**). Probabilities density plot, (**D**). Supervised Principal Component Analysis plot depicting discrimination between AD patients and age–sex matched cognitively healthy individuals, (**E**). Cumulative feature importance plot of the 25 mRNA predictors of the reference signature.

**Figure 4 jcm-09-03016-f004:**
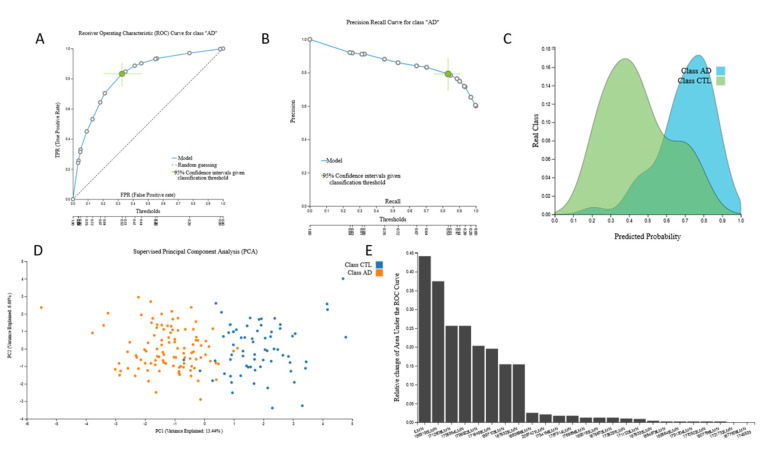
The 38,327-feature mRNA-transcriptomic sub-dataset (train) signature performance, selected predictors and model inspection: (**A**). Receiver Operating Characteristic (ROC) Curve for AD class, (**B**). Precision-recall plot, (**C**). Probabilities density plot, (**D**). Supervised Principal Component Analysis plot depicting discrimination between AD patients and age–sex matched cognitively healthy individuals, (**E**). Cumulative feature importance plot of the 25 mRNA predictors of the signature.

**Figure 5 jcm-09-03016-f005:**
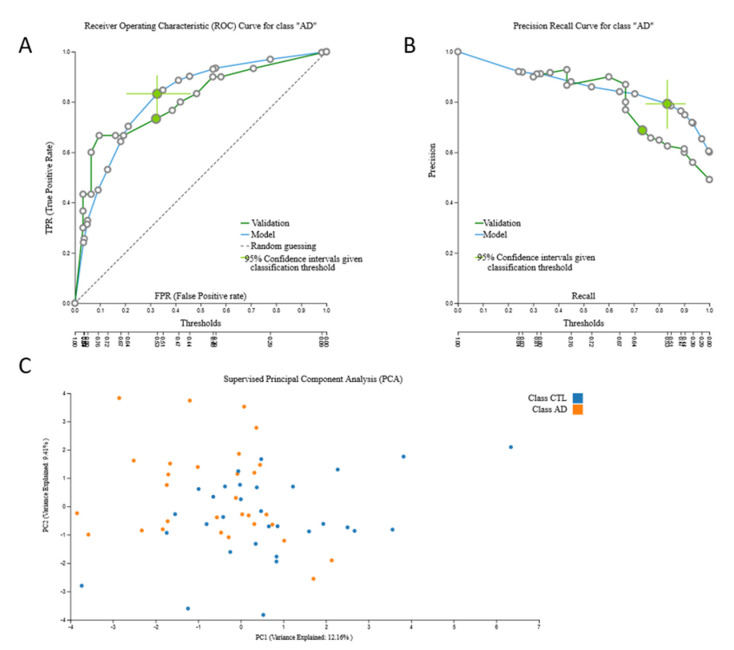
Validation of 38,327-feature mRNA transcriptomic sub-dataset (test) signature performance: (**A**). Receiver Operating Characteristic (ROC) Curve for AD class, (**B**). Precision-recall plot, (**C**). Supervised Principal Component Analysis plot depicting discrimination between AD patients and age–sex matched cognitively healthy individuals.

**Table 1 jcm-09-03016-t001:** Datasets and selected samples used in the study.

Dataset	Data Type	Alzheimer’s Disease Samples	Cognitively Healthy Samples	Features
Metabolomic 1	Metabolites profiles	15	15	3734
Metabolomic 2	Sphingolipid and fatty acid profiles	18	21	25
Proteomic	Protein profiles	25	37	9483
Transcriptomic 1	miRNA profiles	48	22	506
Transcriptomic 2	miRNA profiles	300	289	2566
Transcriptomic 3	mRNA profiles	134	100	38,327
Transcriptomic 4	mRNA profiles	126	131	32,053

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
