# Peer review of "Accurate Blood-Based Diagnostic Biosignatures for Alzheimer’s Disease via Automated Machine Learning"

_jcm, 2020, doi:10.3390/jcm9093016_

Round 1
Reviewer 1 Report
Many thanks for asking me to review this most interesting paper.
My comments come from the clinical perspective, and I will leave comments on the statistical methods and machine learning approaches to others more expert in the field. I comment on the assumption that the statistical approach taken by the authors is considered valid.
From this perspective, there are some areas in the design and conclusions that I think would benefit from some expansion and/or clarification, as outlined below.
One of the issues rightly raised by the authors is that the use of biomarkers would be useful in early diagnosis - otherwise, the diagnosis can be made using current methods. Is there any indication in the underlying datasets of where in the stage of AD these blood samples were taken? In the larger datasets is there information on duration of AD before sampling? Is there any way of linking to disease progression or severity? It may be that data is lacking, or that such approaches would be future work, but a comment on this would be helpful.
The other issue is that clinically, an important issue is not so much differentiating from healthy controls, but distinguishing one type of dementia from another - eg AD from vascular dementia or mixed presentations. Can the authors comment on the utility of their biomarkers in this regard?
On a practical level, how feasible would applying these markers be in a clinical setting? Where would one see their utility, and how feasible (eg scale, cost) is the routine laboratory analysis of the various measures used? Or do the authors envisage this to be more of a tool to aid research studies/clinical trials? Clarification would be useful.
There is also the issue of how one might combine the proteomic, transcriptomic and metabolomic approaches - they seem rather "stand alone" in the current approach. How could one combine them for a better outcome? Have any studies (either in AD or other disorders) used combined -omic measures effectively?
The gold standard for any potential biomarker is to take it from one sample to another one and show that it continues to perform well. As the authors point out, that has not been possible here, albeit that split sample approaches partially address this. One issue raised from this paper is that the different samples give different, albeit overlapping, biosignatures. Can the authors propose a specific signature to test in future samples? Or is the approach one that will constantly evolve as more and more data is gathered and entered into the machine learning protocols? Again, some clarification of the likely pathway to future validation clinical relevance here would be useful.
Author Response
We thank the reviewer for the thorough evaluation of our manuscript and the constructive comments. We addressed the issues raised and made significant additions to our manuscript, that we believe clarify several aspects of the study, improving its comprehension and making it more appealing to a clinical orientated audience. Please find below our specific point-by-point response:
One of the issues rightly raised by the authors is that the use of biomarkers would be useful in early diagnosis - otherwise, the diagnosis can be made using current methods. Is there any indication in the underlying datasets of where in the stage of AD these blood samples were taken? In the larger datasets is there information on duration of AD before sampling? Is there any way of linking to disease progression or severity? It may be that data is lacking, or that such approaches would be future work, but a comment on this would be helpful.
We thank the reviewer for this very reasonable comment. The only available information about the patients/cases in the datasets available in the repositories, also those used in this study, are age and sex (now presented in Supplementary Table 1 of the manuscript). Clinical information is restricted to the presence of diagnosed AD or not (cognitive healthy controls). No information whatsoever is provided on the stage, duration since diagnosis, disease progression, moreover on the type of dementia. This is a most common situation in the deposited –omics datasets which undoubtfully raises an intrinsic limitation of these data-driven approaches. Even more, as re-analyzing and re-evaluating archived data attracts gradually the interest of researchers, as new analyzing tools emerge, such as this used in our study, the quality and completeness of the data stored becomes a major issue and groups producing the original datasets are encouraged to share more information about their samples. This is strongly supported by studies such as this one, aiming to highlight the potential of these approaches.
The following text in now added in the Materials and Methods, l84:
‘The available information about the patients/cases in the datasets used regarding age and sex are presented in Supplementary Table 1.’
The following Table is added in the Supplementary data:
Supplementary Table 1 Available information about the patients/cases in the datasets used regarding age and sex.
Dataset |
|
Alzheimer’s disease samples |
Cognitively healthy samples |
Metabolomic 1 |
Mean Age (y) Gender |
Non available
|
Non available
|
Metabolomic 2 |
Mean Age (y) Gender |
79 4 female/14 male |
82 5 female/16 male |
Proteomic |
Mean Age (y) Gender |
65 16 female/10 male |
69 15 female/22 male |
Transcriptomic 1 |
Mean Age (y) Gender |
70 25 female/23 male |
67 11 female/11 male |
Transcriptomic 2 |
Mean Age (y) Gender |
72 190 female/110 male |
72 137 female/152 male |
Transcriptomic 3 |
Mean Age (y) Gender |
75 91 female/43 male |
73 59 female/41 male |
Transcriptomic 4 |
Mean Age (y) Gender |
77 75 female/51 male |
76 78 female/53 male |
The following clarifying text in now added in the Discussion:
‘The only available information about the patients/cases in the datasets available in the repositories, also those used in this study, are age and sex. Clinical information is restricted to the presence of diagnosed AD or not (cognitive healthy controls). No information whatsoever is provided on the stage, duration since diagnosis, disease progression, moreover on the type of dementia. This is a most common situation in the deposited –omics datasets which undoubtfully raises an intrinsic limitation of these data-driven approaches. Even more, as re-analyzing and re-evaluating archived data attracts gradually the interest of researchers, as new analyzing tools emerge, such as this used in our study, the quality and completeness of the data stored becomes a major issue and groups producing the original datasets are encouraged to share more information about their samples. This is strongly supported by studies such as this one, aiming to highlight the potential of these approaches.’
The other issue is that clinically, an important issue is not so much differentiating from healthy controls, but distinguishing one type of dementia from another - eg AD from vascular dementia or mixed presentations. Can the authors comment on the utility of their biomarkers in this regard?
As discussed above, unfortunately, our biosignatures couldn’t offer any diagnostic information about the different types of dementia, due to the lack of this piece of information in the available datasets. The biosignatures were built in order to differentiate between AD and healthy controls, as this was the only possible clinical end-point. Nevertheless, they offer a mature starting point for researchers working on developing diagnostic tools in AD, should they choose to further check their performance against clinically important end-points such as these pointed by the reviewer, in new and well described datasets, hopefully saving time and resources.
The following clarifying text in now added in the Discussion:
‘The biosignatures built here can only differentiate between AD and healthy controls, as this was the only available clinical end-point. Our analysis cannot offer any additional diagnostic information against other clinically important endpoints, such as the type of dementia, disease progression and therapeutic outcome. Nevertheless, they offer a mature starting point for researchers working on developing diagnostic tools in AD, should they choose to further check their clinical performance in new and well described datasets, hopefully saving time and resources.’
On a practical level, how feasible would applying these markers be in a clinical setting? Where would one see their utility, and how feasible (eg scale, cost) is the routine laboratory analysis of the various measures used? Or do the authors envisage this to be more of a tool to aid research studies/clinical trials? Clarification would be useful.
We thank the reviewer for this comment that give us the opportunity to clarify this important aspect of our work: upon further validation in complete datasets by groups with a special interest in AD, these biosignatures could offer feasible solutions for routine blood-based laboratory tests that could be realized in any standard equipped diagnostic lab. This is the most significant merit of the present study for the scientific community, moving from the –omics multi-dimensional results to simpler classifiers. Reducing the dimensions of a signature is a great advantage in terms of translationability to cost-effective assays with lesser need of multiplexing. In specific, proteomic and miRNA biosignatures could be applied in a clinical setting due to their limited number of predictors. Proteomic biosignature could be used as a 7-protein ELISA assay, while miRNA biosignature as a 3-plex RT-PCR assay measured on plasma samples, both technically feasible.
The following text in now added in the Discussion:
‘Upon further validation in complete datasets by groups with a special interest in AD, the biosignatures described could offer feasible solutions for routine blood-based laboratory tests that could be realized in any standard equipped diagnostic lab. This is the most significant merit of the present study for the scientific community, moving from the –omics multi-dimensional results to simpler classifiers. Reducing the dimensions of a signature is a great advantage in terms of translationability to cost-effective assays with lesser need of multiplexing. In specific, proteomic and miRNA biosignatures could be applied in a clinical setting due to their limited number of predictors. Proteomic biosignature could be used as a 7-protein ELISA assay, while miRNA biosignature as a 3-plex RT-PCR assay measured on plasma samples, both technically feasible’.
There is also the issue of how one might combine the proteomic, transcriptomic and metabolomic approaches - they seem rather "stand alone" in the current approach. How could one combine them for a better outcome? Have any studies (either in AD or other disorders) used combined -omic measures effectively?
As aptly pointed by the reviewer, the available datasets do not jointly include all modalities on the same samples. Clearly, as these are laborious and costly tests, it is rare to employ more that one in a single study sample lot. Thus, it is impossible for JADBIO or any other tool to analyze them in combination. There are two ways to combine these data, as far as we know:
- Measure all modalities on the same sample lot in a new, multi-omics study. This is the standard approach taken by some advanced technology labs, with the obvious downside of requiring new and expensive studies. We were not able to identify any such dataset for AD, including all three platforms or any combination of two.
- Create computational methods that "translate" one platform and technology (e.g., transcriptomic) to another (e.g., proteomics). At the moment, this is at the spearhead of bioinformatics development, with several labs (including ours) intensively working on such technologies, however still with under-optimal results (Normand et al, 2018).
Normand, R., Du, W., Briller, M. et al. Found In Translation: a machine learning model for mouse-to-human inference. Nat Methods 15, 1067–1073 (2018). https://doi.org/10.1038/s41592-018-0214-9
Nevertheless, as far as it concerns applicability as an effective test in a clinical setting, biosignatures combining features of different nature (i.e. both nucleic acids and proteins) would be technically more complicated to implement in a single blood sample.
The following text in now added in the Discussion:
‘Combining proteomic, transcriptomic and metabolomic data in a multi-omic approach could theoretically be implemented to more effective data exploitation. However, any datasets jointly including more that one modalities on the same sample set are rare, obviously due to high resources requirements. Alternatively, efforts to create computational methods that "translate" one platform and technology (e.g., transcriptomic) to another (e.g., proteomics) are the moment the spearhead of bioinformatics development, with several labs (including ours) intensively working on such technologies, however still with under-optimal results [32]. Nevertheless, as far as it concerns applicability as an effective test in a clinical setting, biosignatures combining features of different nature (i.e. both nucleic acids and proteins) would be technically more complicated to implement in a single blood sample.’
The gold standard for any potential biomarker is to take it from one sample to another one and show that it continues to perform well. As the authors point out, that has not been possible here, albeit that split sample approaches partially address this. One issue raised from this paper is that the different samples give different, albeit overlapping, biosignatures. Can the authors propose a specific signature to test in future samples? Or is the approach one that will constantly evolve as more and more data is gathered and entered into the machine learning protocols? Again, some clarification of the likely pathway to future validation clinical relevance here would be useful.
Indeed, enrichment of the repositories with new –omics datasets would possibly lead in the future to more accurate biosignatures. However, as these platforms have already been used for this particular disease, it is rather unlikely that we will soon have more such data, given also the increased resources required. We anticipate that the momentum is mature for extracting meta-analysis information, especially as we now have advanced bioinformatics tools such as JADBIO, capable of processing low-sample high dimensional datasets to optimal conclusions.
All three reported biosignatures showing high performance and AUC>0.85 are worthy of further validation. Still, if we were to choose one as the most promising, we would propose this to be the miRNA-transcriptomic biosignature based on three apparent advantages: highest performance (AUC 0.975 [0.906, 1.000]), less features (three -predictors), entities allowing easy multiplexing assays (miRNA).
The following texts are now added in the Discussion:
‘A side-conclusion of this study, yet quite important in our opinion, is the following: revisiting and reanalyzing "old" datasets is not only potentially fruitful but scientifically necessary. The current mentality of life-scientists is often to discard public data as "used", under the assumption that all discoveries to be made, are already published. However, as new and powerful statistical and computational methods are introduced, such as JADBIO and AutoML, new types of analyses become possible and new patterns and results become ripe for discovery[33]. It is not only computational methods that change; the analysis context is constantly changing. It is thus worth re-analyzing past datasets in the context of new datasets submitted [24], recent scientific publications, or newly added or revise knowledge in biological databases (e.g., updated pathways).’
‘All three reported biosignatures showing high performance and AUC>0.85 are worthy of further validation. Still, if we were to choose one as the most promising, we would propose this to be the miRNA-transcriptomic biosignature based on three apparent advantages: highest performance (AUC 0.975 [0.906, 1.000]), less features (three -predictors), entities allowing easy multiplexing assays (miRNA).’

Reviewer 2 Report
The aim of this cross sectional study is to identify diagnostic biosignatures that distinguish AD patients from health congnitvely normal subjects. The authors utilized publicly available high-throughput low-sample -omics datasets for automated machine learning (autoML), using the Just Add Data Bio (JADBIO) platform.
The authors identify three best performing diagnostic biosignatures: a) a 506-feature transcriptomic dataset (AUC 0.975 [0.906, 1.000]), b) a 38,327-feature transcriptomic (AUC 0.846 [0.778, 0.905]) and c) a 9483-feature proteomic (AUC 0.921 [0.849, 0.972]).
I have many difficulties in to review this paper. I suggest to improve the understanding of this paper also for a clinical expert in dementia.
Here are the my suggestions:
- The authors do not report (if available in the pubblicly data-set) clinical and demographical information on AD patients and controls (i.e. type and severity of dementia, sex, age, education). This is crucial to replicate these data in an other scientific report.
- The AUC approach allows to identify the best cut-off of a possible diagnostic test. The authors should identify for these three possible biomarkers a cut-off and to calculate sensitivity, specificity, negative and positive predicitive values. This clinical and epidemiological approach allows to understand better the clinical utility of these possibile biomarkers.
- The biological mechanism of these possible biomarkers should be reported and deeply discussed.
- The authors should better clarify the internal and external validation process using bootstrapping techniques
Author Response
We thank the reviewer for the thorough evaluation of our manuscript and the constructive comments. We addressed the issues raised and made significant additions to our manuscript, that we believe clarify several aspects of the study, improving its comprehension and making it more appealing. Please find below our specific point-by-point response (see also attached file):
- The authors do not report (if available in the publicly data-set) clinical and demographical information on AD patients and controls (i.e. type and severity of dementia, sex, age, education). This is crucial to replicate these data in another scientific report.
The only available information about the patients/cases in the datasets available in the repositories, also those used in this study, are age and sex (now presented in Supplementary Table 1 of the manuscript). Clinical information is restricted to the presence of diagnosed AD or not (cognitive healthy controls). No information whatsoever is provided on the stage, duration since diagnosis, disease progression, moreover on the type of dementia. This is a most common situation in the deposited –omics datasets which undoubtfully raises an intrinsic limitation of these data-driven approaches. Even more, as re-analyzing and re-evaluating archived data attracts gradually the interest of researchers, as new analyzing tools emerge, such as this used in our study, the quality and completeness of the data stored becomes a major issue and groups producing the original datasets are encouraged to share more information about their samples. This is strongly supported by studies such as this one, aiming to highlight the potential of these approaches.
The following text in now added in the Materials and Methods, l84:
‘The available information about the patients/cases in the datasets used regarding age and sex are presented in Supplementary Table 1.’
The following Table is added in the Supplementary data:
Supplementary Table 1: Available information about the patients/cases in the datasets used regarding age and sex.
Dataset |
|
Alzheimer’s disease samples |
Cognitively healthy samples |
Metabolomic 1 |
Mean Age (y) Gender |
Non available
|
Non available
|
Metabolomic 2 |
Mean Age (y) Gender |
79 4 female/14 male |
82 5 female/16 male |
Proteomic |
Mean Age (y) Gender |
65 16 female/10 male |
69 15 female/22 male |
Transcriptomic 1 |
Mean Age (y) Gender |
70 25 female/23 male |
67 11 female/11 male |
Transcriptomic 2 |
Mean Age (y) Gender |
72 190 female/110 male |
72 137 female/152 male |
Transcriptomic 3 |
Mean Age (y) Gender |
75 91 female/43 male |
73 59 female/41 male |
Transcriptomic 4 |
Mean Age (y) Gender |
77 75 female/51 male |
76 78 female/53 male |
The following clarifying text in now added in the Discussion:
‘The only available information about the patients/cases in the datasets available in the repositories, also those used in this study, are age and sex. Clinical information is restricted to the presence of diagnosed AD or not (cognitive healthy controls). No information whatsoever is provided on the stage, duration since diagnosis, disease progression, moreover on the type of dementia. This is a most common situation in the deposited –omics datasets which undoubtfully raises an intrinsic limitation of these data-driven approaches. Even more, as re-analyzing and re-evaluating archived data attracts gradually the interest of researchers, as new analyzing tools emerge, such as this used in our study, the quality and completeness of the data stored becomes a major issue and groups producing the original datasets are encouraged to share more information about their samples. This is strongly supported by studies such as this one, aiming to highlight the potential of these approaches.’
‘The biosignatures built here can only differentiate between AD and healthy controls, as this was the only available clinical end-point. Our analysis cannot offer any additional diagnostic information against other clinically important endpoints, such as the type of dementia, disease progression and therapeutic outcome. Nevertheless, they offer a mature starting point for researchers working on developing diagnostic tools in AD, should they choose to further check their clinical performance in new and well described datasets, hopefully saving time and resources.’
- The AUC approach allows to identify the best cut-off of a possible diagnostic test. The authors should identify for these three possible biomarkers a cut-off and to calculate sensitivity, specificity, negative and positive predictive values. This clinical and epidemiological approach allows to understand better the clinical utility of these possible biomarkers.
The optimal cut-off threshold for clinical use depends on two factors: (a) the cost (not only the monetary cost but cost in a general sense) of misclassifying an Alzheimer's case to a healthy (false negative) and vice versa (false positive) and (b) the prevalence of Alzheimer's syndrome to the population where the model is applied. An optimal threshold cannot be selected without these two pieces of information.
However, JADBIO facilities the selection of a clinical cut-off threshold by providing the ROC curve, which contains all possible trade-offs between the False Positive Rate and the True Positive Rate, as well as the Precision-Recall Curve which contains all possible trade-offs between the Precision and Recall that are achievable by the model. JADBIO calculates statistics, performance metrics, and confidence intervals for 10 such points on these two curves.
As a response to the reviewer’s comments, we now included this text in the Results (line 150):
‘Selection of a clinical cut-off threshold taking into consideration cost and disease prevalence is facilitated by JADBIO ROC curve outcome, which contains all possible trade-offs between the False Positive Rate and the True Positive Rate, as well as the Precision-Recall Curve which contains all possible trade-offs between the Precision and Recall that are achievable by the model. JADBIO calculates statistics, performance metrics, and confidence intervals for 10 such points on these two curves (Figure 2 A and B). Performance metrics (sensitivity, specificity, PPV, NVP, etc.) for three standard cut-off thresholds of this particular biosignature are included in Supplementary Figures 7, 8 and 9.
Further interactive peruse for cut-off thresholds and metrics for this analysis is provided in the link https://app.jadbio.com/share/3f050861-da6b-447a-b2b1-908c71d65c3d.’
Furthermore, we added the following figures in the Supplementary Materials, which describe the performance metrics (sensitivity, specificity, PPV, NVP, etc.) for three cut-off thresholds:
Supplementary figure 7. Performance metrics when classification threshold (0.440) is optimized for best Accuracy.
Supplementary figure 8. Performance metrics when classification threshold (0.981) is optimized for best Specificity.
Supplementary figure 9. Performance metrics when classification threshold (0.197) is optimized for best True positive rate.
- The biological mechanism of these possible biomarkers should be reported and deeply discussed.
We thank the reviewer for this comment. Coming from a biomedical background, we too were deeply engaged into the discovery of the biological significance of our findings and the role of each one of the identified features in the disease pathophysiology. We draw the attention of the reviewer to the lines 292-304 of the discussion, where we summarize the result of this work related to the protein-based biosignatures, concluding that they mostly present biomarkers so far neglected in AD studies. Still, we soon realized that most of the identified features presented no obvious or strong connection with AD and trying to build a regulatory hypothesis based simply in the fact that they contribute to a classification algorithm would not be substantiated enough, would require much further experimentation and would translocate the focus of our paper from its main objective: to revisit published precious –omics datasets via AutoML tools which only recently became available to produce new reduced-feature models that could subsequently be translated to cost-effective benchmark diagnostic solutions.
As a response to the reviewer’s comment, we now included the results from our search using the GeneCards- The Human gene database- search tool. See additions in:
- Materials and Methods:
2.3 Correlation of selected features to AD.
The biological involvement and related pathways of identified features, i.e. proteins, mRNAs and miRNAs to AD was searched using the GeneCards-The Human gene database tool (https://www.genecards.org/). MiRNA Predicted targets were identified in miRbase database (http://www.mirbase.org/).
- Results 3.2.1: Selected protein features include: Leucine Rich Repeats and IQ Motif Containing Protein 2 (LRRIQ2), Calcium signal-modulating cyclophilin ligand (CAMLG), interleukin 4 (IL4), tropomyosin 1 (TPM1), interleukin 20 (IL20), diablo homolog (Drosophila) (DIABLO), and Serine/threonine-protein kinase 3 (VRK3). Relation to AD according to GeneCards search (https://www.genecards.org/Search/Keyword?queryString=Alzheimer%27s&startPage=0&pageSize=-1), revealed some relation to AD for all but the first two proteins, IL4 being the one presenting the highest score (Supplementary Table 2).
- Results 3.2.2: ‘Relation of the 3 identified miRNAs to the AD according to GeneCards search (https://www.genecards.org/Search/Keyword?queryString=ALZHEIMER&pageSize=-1&startPage=0) revealed relation to the disease, with MIR29C related to the MicroRNAs in cancer and Metastatic brain tumor pathway presenting the highest score (Supplementary Table 3). MIR30D and MIR182 were identified in the miRNA targets in ECM and membrane receptors and MicroRNAs in cancer and Alzheimer’s Disease pathways, respectively.’
- Results 3.2.3: ‘Relation of the 30 identified mRNAs to the AD according to GeneCards search (https://www.genecards.org/Search/Keyword?queryString=ALZHEIMER&pageSize=-1&startPage=0) revealed no or poor relation, with the exception of CHAT gene of Choline O-Acetyltransferase, related to the pathways of Neurotransmitter Release Cycle and Transmission across Chemical Synapses ((Supplementary Table 4).
- Discussion l275-276: ‘These three miRNAs were also shown to be related to AD in Genecards search, pointing into a biological role of the pathways involved in disease pathophysiology.’
- Three Tables were included in the Supplementary data, describing the result of the Genecards search for the features selected in the three biosignatures, i.e. 3 miRNAs, 30 mRNAs and 6 proteins.
- The authors should better clarify the internal and external validation process using bootstrapping techniques
The bootstrapping technique used performs a correction to the estimation of out-of-sample performance of the final model. The correction (adjustment) is required because JADBIO tries thousands of machine learning pipelines to identify the best one that produces the optimal, final model. The correction is conceptually similar to the Bonferroni adjustment required for multiple hypotheses testing due to performing multiple tests. Intuitively, the selection process, which selects the best out of numerous pipelines is bootstrapped. This technique has been shown to produce conservative estimates of performance in massive evaluation experiments with general types of data [see Ref Tsamardinos et al., 2018] as well as hundreds of -omics data [see ref Kerkentzes et al., 2014]. It has been used to produce several novel scientific results [see Refs 9-12].
- Kerkentzes, V. Lagani, I. Tsamardinos, M. Vyberg, and O. Røe, "Hidden treasures in “ancient” microarrays: gene-expression portrays biology and potential resistance pathways of major lung cancer subtypes and normal," Frontiers 2014, iss. 251, 2014. doi:10.3389/fonc.2014.00251
Tsamardinos I, Greasidou E, Borboudakis G. Bootstrapping the out-of-sample predictions for efficient and accurate cross-validation. Mach Learn [Internet]. 2018/05/09. 2018;107(12):1895–922. Available from: https://pubmed.ncbi.nlm.nih.gov/30393425
The following clarifying text in now added in the Discussion:
‘The bootstrapping technique used performs a correction to the estimation of out-of-sample performance of the final model. The correction (adjustment) is required because JADBIO tries thousands of machine learning pipelines to identify the best one that produces the optimal, final model. The correction is conceptually similar to the Bonferroni adjustment required for multiple hypotheses testing due to performing multiple tests. Intuitively, the selection process, which selects the best out of numerous pipelines is bootstrapped. This technique has been shown to produce conservative estimates of performance in massive evaluation experiments with general types of data [23] as well as hundreds of -omics data [24]. It has been used to produce several novel scientific results [see Refs 9-12].’
Also:
‘A side-conclusion of this study yet quite important, is the following: revisiting and reanalyzing "old" datasets is not only potentially fruitful but scientifically necessary. The current mentality of life-scientists is often to discard public data as "used", under the assumption that all discoveries to be made, are already published. However, as new and powerful statistical and computational methods are introduced, such as JADBIO and AutoML, new types of analyses become possible and new patterns and results become ripe for discovery [33] It is not only computational methods that change; the analysis context is constantly changing. It is thus worth re-analyzing past datasets in the context of new datasets submitted [24] recent scientific publications, or newly added or revise knowledge in biological databases (e.g., updated pathways).’

Round 2
Reviewer 2 Report
I think that paper is much improved.